REGISTERED REPORT PROTOCOL

# Parents' awareness, knowledge, and experiences of play and its benefits in child development: A systematic review protocol

**Brightlin Nithis Dhas**[1]*, **Shoba Mary Chacko**[1], **Vince Soloman David Solomon**[1], **Vimal Sriram**[2]

**1** Department of Occupational Therapy, Hamad Medical Corporation, Doha, Qatar, **2** Collaborative Learning and Capacity Building Theme, NIHR ARC NWL, London, United Kingdom

* bdhas@hamad.qa

## Abstract

### Background

Play is an important childhood occupation and a medium for development. Parents' attitudes towards play, knowledge about play and its benefits, and their experiences in facilitating effective play are key factors that determine the experiences of play in children. These factors related to parent's understanding and experiences of play gain additional significance when the child has a disability. The aim of this systematic review is to synthesize the available evidence on awareness, knowledge, and experience of play among parents and to summarise the findings.

### Method

MEDLINE, CINAHL, APA PsycINFO, Psychology Database, Sociological Abstracts, EMBASE, and Cochrane Collection Plus will be searched for studies of any design that investigate awareness, knowledge, and experience of play among parents and its benefits to child development. Manual searches from reference lists of relevant papers will also be completed. The primary outcomes are parents' knowledge (what constitutes play), experiences (what parents feel about play) and awareness (benefits of play) about play. Three independent reviewers will screen identified papers with pre-defined eligibility criteria and extract data using a customized extraction form. Discrepancies will be resolved in discussion with a fourth reviewer. A synthesis of eligible studies and summary will be provided.

### Discussion

A systematic review of quantitative and qualitative research evidence of parents' awareness, knowledge, and experiences in play will be carried out. This will highlight parents' own views on play among their children, current theories/domains related to parents' awareness, knowledge, and experience in play, and outcome measures that have been used. In addition, comparison among views of parents of children with disabilities and parents of typically

**Data Availability Statement:** All relevant data from this study will be made available upon study completion.

**Funding:** Hamad Medical Corporation provided support in the form of salaries for authors BND, SMC and VSDS. The specific roles of these authors are articulated in the 'author contributions' section. The funders had no role in study design, data collection and analysis, decision to publish, or preparation of the manuscript.

**Competing interests:** The authors have read the journal's policy and have the following competing interests: BND, SMC and VSDS are paid employees of Hamad Medical Corporation and VS is a paid employee of NIHR Applied Research Collaboration. The authors declare that this article presents independent research commissioned by the National Institute for Health Research (NIHR) under the Applied Health Research (ARC) programme for Northwest London. The views expressed in this publication are those of the author(s) and not necessarily those of the NHS, the NIHR or the Department of Health. This does not alter our adherence to PLOS ONE policies on sharing data and materials. There are no patents, products in development or marketed products associated with this research to declare.

**Abbreviations:** CINAHL, Cumulative Index of Nursing and Allied Health Literature; COSMIN, Consensus-based Standards for the selection of health Measurement Instruments; PRISMA–P, Preferred Reporting Items for Systematic Reviews and Meta-Analyses Protocols; PROSPERO, International Prospective Register of Systematic Reviews.

developing children will be made. The results will be presented as a summary of key findings under the themes of awareness, knowledge, and experience of parents in play.

## Introduction

Play has tremendous growth promoting potential in a range of developmental domains including cognition, language, social integration, social communication, and emotional regulation [1, 2]. Children have an innate desire to play and through play they develop many life skills without even realizing it [3]. Everyone acknowledges that play is an important aspect of child development. However, there are considerable differences among parent and professional conceptualisations of play and its value to child development [4]. In their study, Fisher et al. [4] found that mothers consider both structured activities (goal-oriented activities consisting of a sequence of events/actions) and unstructured activities (highly varied activities involving imagination/creativity) as highly playful, whereas, professionals' rate structured activities to be less playful. With regards to its developmental value, mothers felt unstructured activities contribute less to child development compared to professionals, while both parents and professionals ascribed high developmental value to structured activities.

There are many factors that influence play, among them, family is an important factor [5, 6]. Parental beliefs about play are found to affect the play experiences of their children [7]. Parents are considered to have a "gatekeeper" role in determining which activities children do and therefore their awareness towards the importance of play in child development can have an impact on children's play opportunities. Parents who value the importance of play devote more time for play [3] and encourage the type of play they deem to be important [8]. For example, parents who place importance on physical activities tend to provide more opportunities for physical play while those who place importance on intellectual activities tend to provide more opportunities for intellectual play [3].

Knowledge regarding play among parents is another factor in determining play experiences of their children. For example, what toys to buy based on the developmental level of the child is important as this could determine if the child derives enjoyment from playing with them or the child feels frustrated because they cannot play with it [3]. Children whose parents had better knowledge about play were likely to engage in higher level of play [9]. Certain disabling conditions impose specific limitations on playing skills for children with these conditions [10]. For example, play participation in children with Cerebral Palsy was linked to their motor skills [11] while for children with Autism, certain play skills are impaired [12]. Knowledge about overcoming these obstacles through changes or adaptations to toys and or environment would make a difference in play participation for their children [13]. Even if parents were aware and knowledgeable about play and its benefits, it does not translate into them being competent enough to play with their children effectively [14]. Therefore, it is suggested that parents need some training on how to play with their children in order to make their play interactions rewarding and already some positive results has been observed on interventions targeting teaching play skills to parents [15].

Studies on parental experiences while playing with their children that explore questions related to "how, what, when, where and why" parents play with their child provide valuable insight on the level and type of parental engagement in their children's play [16]. The amount of time parents play with their children, the type of engagement during play and the toys/play materials they buy and use with their children depends on their viewpoints about play [17].

Play does not exist equally for all children but determined by biological factors such as gender and developmental transitions, socio-cultural factors, and specific context such as presence of an illness or disability [18]. Therefore, there could be variations in the awareness, knowledge, and experiences among parents of children with different developmental levels, culture, and children with physical, social and intellectual disabilities. For this review, we will examine studies reporting parent's awareness, knowledge, and experience with play among children and describe the variations in terms of age groups and disability status of the children, gender of parents, and the cultural context (where reported).

## Rationale for the review

Parents are an integral part of their children's play and the play experiences are closely associated with the choices that parents make [19, 20]. Therefore, parents' understanding of the importance or nature of play is essential in creating nurturing play experiences for children. Particularly for children with disabilities whose engagement in play occupations could be limited due to impairments in play skills, environmental barriers, or both. Parent knowledge on developmental aspects of play and adapting play could thus make a difference. Currently, there are several studies exploring the positive benefits of play in child development and interventions to improve play performance on children [21, 22]. However, little is known about how parents feel about play and if they perceive that they have enough knowledge and competence in structuring play experiences for their children. This review aims to appraise the literature on play from parents' perspective in order to synthesize the available evidence in this important field of enquiry.

## Review aims and questions

This review will aim to:

1. Synthesize qualitative, quantitative, and mixed methods research related to awareness, knowledge, and experience in play among parents of typically developing children.

2. Synthesize qualitative, quantitative, and mixed methods research related to awareness, knowledge, and experience in play among parents of children with disabilities.

3. Compare 1 and 2 to identify similarities and differences in conceptualization of play by parents of typically developing children and those having children with disabilities.

The following information will also be collected and reported: Details of the participants in the reviewed studies; surveys/observational methods/standardized tools used in understanding parents' awareness, knowledge and experience in play and interventions targeting knowledge and competence of parents in play.

## Methods/Design

This protocol has been prepared following the Preferred Reporting Items for Systematic Review and Meta-Analysis Protocols (PRISMA-P) guidelines for systematic review protocols [23]. (The PRISMA-P checklist is included as S1 File). The review protocol has been registered with the International Prospective Register of Systematic Reviews PROSPERO (CRD42021259601).

## Inclusion/Exclusion criteria

**Types of participants.**   Studies that include parents, of typically developing children or children who have any type of disabilities will be included. There will be no restrictions

regarding sex, medical condition of the child or ethnic background. Studies that report on parents whose children have reached adolescence (above 11 years of age) will be excluded.

**Types of studies.** Studies that focus on exploring or improving parents' understanding of play will be included. Both quantitative and qualitative study designs are required to answer the review questions. Hence no restrictions will be placed on study design. The full-text article of the study needs to be available (i.e. abstracts will be excluded). We will not be including other reviews; however, we will check references within identified existing reviews on parents' awareness, knowledge, and experiences of children's play to ensure that all relevant studies have been located. Non peer-reviewed publications including letters to the editor, abstract and conference proceedings, magazine articles, and book reviews will not be included. Study protocols and theses/dissertations will also be excluded.

**Types of play.** Since there is no universal definition of play, we will use a broad definition and all research articles that consider play in general (not specific play types such as digital play, pretend play, etc.), done with the purpose of enjoyment and not for competition (sports), achievement (physical activity or fitness), or therapeutic purpose (play therapy) will be included. The focus of many studies is on improving play skills among children or using play as a therapeutic media. However, this review's focus is on parent perceptions of play and interventions directed towards parents to address their concerns. We will report on the common definitions and interpretations of their children's play from the perspectives of parents.

**Comparator(s).** Perspectives of parents of typically developing children will be compared to the experiences of parents of children with disabilities.

**Outcome measures.** The primary outcomes will be the level of parents' awareness, knowledge, and experiences of play. Awareness refers to the level of importance parents' attach to play in the development of the child. Knowledge covers understandings of the concept of play, its types, and the developmental stages of play. Experience relates to how, what, when, where and why parents play with their children and how they organize their schedules, objects, and environments to facilitate play.

Secondary outcomes will be to identify any survey/measures or instruments developed for assessing parents' awareness, knowledge, and experience in relation to their children's play. We will also identify and report on any changes in trends of how parents' awareness, knowledge and experience of children's play has changed over time.

## Search strategy

Searches will be carried out on:

1. Databases

In order to carry out a comprehensive review, we will search for relevant studies in more than one database. The databases of CINAHL (EBSCO) from 1937 to present, MEDLINE (EBSCO) from 1946 to present; APA PsycINFO (ProQuest) from 1806 to present; Psychology Database (ProQuest) from inception to present; Sociological Abstracts (ProQuest) EMBASE from 1974 to present; and Cochrane Collection Plus (EBSCO).

2. Manual searches

Manual searches of reference lists and included articles in any previous reviews or metanalyses will also be conducted to identify relevant research studies.

Searches will not be limited by quantitative or qualitative filters or date limits. However, the search will be limited to articles in English language due to funding constraints.

The search strategy for the search in Medline (EBSCO) is provided (S2 File).

### Screening

The web based Covidence software platform (https://www.covidence.org/) will be used for the review. Search results from the databases will be downloaded into Covidence software. Covidence will be used to identify and remove duplicates. Authors BND, VSDS and SMC will screen all titles and abstracts independently based on the inclusion/exclusion criteria. Any disagreement will be resolved by discussion and in consensus with VS. Full text review will then be conducted for all studies marked for inclusion. An audit trail will be maintained for exclusion of any studies at this stage by adding notes in Covidence.

### Data extraction

A bespoke form (S3 File) developed by the authors will be used for data extraction. The subheadings in the form were taken from relevant sections from the STROBE checklist (The Strengthening the Reporting of Observational Studies in Epidemiology) [24] and the COREQ checklist (Consolidated criteria for reporting qualitative research) [25]. This form will be piloted on a sample of studies initially and modified if needed. BND, SMC and VSDS will extract the data.

### Risk of bias (quality) assessment

The Mixed Methods Appraisal Tool [Version 18] [26] for assessing bias in both quantitative and qualitative studies will be used. The COSMIN Study Design checklist for Patient-reported outcome measurement instruments [version July 2019] [27] will be used for assessing any surveys developed to assess parents' understanding of play. The selected studies will be critically evaluated by BND, VSDS and SMC and any discrepancies resolved in discussion with VS.

### Data synthesis

A descriptive summary of included studies will be created with a meta-analysis if the included studies are sufficiently homogenous [28]. If the studies are not homogenous a narrative synthesis will be used. An applied thematic analysis approach [29] will be used to synthesize findings from qualitative studies. Where data is available, a subgroup analysis based on parents of typically developing children and parents of children with disabilities will be conducted.

### Potential limitations

Certain limitations are anticipated. Since there are numerous terminologies used to describe play and parents' understanding of it some relevant studies could potentially be missed and heterogeneity in the included studies could be expected. Restricting the searches to limited databases and limiting to only English language peer-reviewed articles is another limitation as some important articles could be missed. However, we are confident that our comprehensive search strategy, searches in multiple databases, manual search among reference lists and using a broad definition of play will provide a comprehensive review in this important topic area.

## Discussion

The aim of this review is to understand the level of awareness, knowledge, and experience of play among parents. The results are expected to summarize parents' perceptions of various dimensions of play including importance of play, attitudes towards play, and experience in supporting play. Existing theoretical perspectives of play could be compared with the parent perspectives derived from the review. This systematic review would also help to identify any lack of awareness or negative attitudes towards play from the part of parents that needs to be

addressed as parental views about play affects actual play among children which in turn has the potential to affect child development [7, 19, 20]. Gaps in the literature, if any in the understanding of parental concerns and competence in playing with their children, particularly children with disabilities could be identified from the review. Moreover, comparisons of viewpoints of parents who have children with disabilities and those of typically developing children could provide insights regarding the complexities, hardships if any, and unique experiences parents of children with disabilities face while playing with their children. In addition, the review will also identify methods used in the literature to identify parental perceptions about play and any standardized questionnaires developed in this regard. This would be useful to identify existing outcome measures in this area. The inclusion of quantitative and qualitative study designs will be helpful to thoroughly explore the available literature related to the aim of this review. It is anticipated that the findings will be useful to develop insights regarding empowering parents to nurture growth of their children through play. Furthermore, areas that require further investigation and research could be identified.

## Supporting information

**S1 File. PRISMA-P checklist.**
(DOCX)

**S2 File. Search strategy.**
(DOCX)

**S3 File. Data extraction forms.**
(DOCX)

## Author Contributions

**Conceptualization:** Brightlin Nithis Dhas, Shoba Mary Chacko, Vimal Sriram.

**Investigation:** Brightlin Nithis Dhas, Shoba Mary Chacko, Vince Soloman David Solomon.

**Methodology:** Brightlin Nithis Dhas, Vimal Sriram.

**Project administration:** Brightlin Nithis Dhas, Shoba Mary Chacko, Vince Soloman David Solomon.

**Writing – original draft:** Brightlin Nithis Dhas.

**Writing – review & editing:** Shoba Mary Chacko, Vince Soloman David Solomon, Vimal Sriram.

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
