## [Decision Letter · Decision Letter 0]

15 Jun 2022

PONE-D-21-25685Parents’ awareness, knowledge, and experiences of play and its benefits in child development: a systematic review protocolPLOS ONE

Dear Dr. Dhas,

Thank you for submitting your manuscript to PLOS ONE. After careful consideration, we feel that it has merit but does not fully meet PLOS ONE’s publication criteria as it currently stands. Therefore, we invite you to submit a revised version of the manuscript that addresses the points raised during the review process.

When revising your manuscript please ensure your address fully the reviewers' comments regarding the selection criteria, in particular those regarding the difference in play between infants and adolescents, and the timescale of the articles to be included given changes in parenting behavior over time.

We look forward to receiving your revised manuscript.

Kind regards,

Hugh Cowley

Staff Editor

PLOS ONE

Journal Requirements:

2. Thank you for stating the following in your Competing Interests section:  "NO authors have competing interests"

5. Thank you for stating the following in the Financial Disclosure section: "The authors received no specific funding for this work."

We note that one or more of the authors are employed by a commercial company: Hamad Medical Corporation

7. We note that this manuscript is a systematic review or meta-analysis; our author guidelines therefore require that you use PRISMA guidance to help improve reporting quality of this type of study. Please upload copies of the completed PRISMA checklist as Supporting Information with a file name “PRISMA checklist”.

Reviewers' comments:

Reviewer's Responses to Questions

**Comments to the Author**

1. Does the manuscript provide a valid rationale for the proposed study, with clearly identified and justified research questions?

Reviewer #1: Yes

Reviewer #2: Yes

2. Is the protocol technically sound and planned in a manner that will lead to a meaningful outcome and allow testing the stated hypotheses?

Reviewer #1: Partly

Reviewer #2: Partly

3. Is the methodology feasible and described in sufficient detail to allow the work to be replicable?

Reviewer #1: Yes

Reviewer #2: Yes

4. Have the authors described where all data underlying the findings will be made available when the study is complete?

Reviewer #1: No

Reviewer #2: No

5. Is the manuscript presented in an intelligible fashion and written in standard English?

Reviewer #1: Yes

Reviewer #2: Yes

6. Review Comments to the Author

You may also provide optional suggestions and comments to authors that they might find helpful in planning their study.

Reviewer #1: The manuscript presents a registered report protocol for a systematic review on parents’ awareness, knowledge, competence and experiences of play. In addition, the authors aim to compare these between parents of children with disabilities and parents of typically developing children. The proposed study is original and should be of interest for the field. Although it is widely accepted that play has a major role in human development, the influence of parental variables has often been neglected in this research field.

I have a few general comments for the authors that I feel should be addressed before the manuscript is further considered for peer review:

Introduction

In general, although very briefly described, the theoretical background presented does point well towards the main research question. However, I think that some more general descriptions of the main concepts would be helpful to lay the foundation for the Methods section. For example, it remains unclear what the authors refer to as “play” and which developmental stage the authors refer to. It is quite obvious that the quality and quantity of children’s play (and thus also parents attitudes towards and their roles in it) will change dramatically throughout development. This is has to be described and the authors have to find a way to account for these developmental transitions in their study, either by focusing on a specific developmental stage or by taking “age” into account as a study variable. Furthermore, the introduction mentions parent’s awareness, knowledge, and competence as the main outcome variables. However, in the methods section (and abstract) it seems that four variables are included: awareness, knowledge, competence and experience. The title, on the other hand, only mentions “awareness, knowledge, and experiences”. This should be resolved and the introduction should mention and explain the main study variables in more detail. The introduction also lacks a particular motivation for the comparison between parents of children with disabilities (again: what do the authors mean here?) and parents of typically developing children.

Methodology

The description of participants states that parents with “children” above 18 years of age will be excluded. Despite the fact that a 17-year-old is barely referred to as a “child” (but rather as an adolescent), the theoretical background presented seems to be focused on children’s play, and not of adolescents. Thus, the authors have to explain why studies with adolescents should be included in the literature review and, if so, have to adapt the theoretical background in order to capture the developmental transition of play from infancy to adulthood.

Some more minor comments:

1. The types of studies included does not exclude previous reviews or meta-analyses – how do the authors go about these studies?

2. What is the reason to search in five different databases and to have three different independent reviewers?

3. The authors state as a limitation that “...since there are numerous terminologies used to describe play and parents’ understanding of it some relevant studies could potentially be missed”. This is why it is important that the authors state their conceptualization of “play” in the introduction and explain in the Methods section how they apply this definition in the literature search.

I hope that the above suggestions help the authors to improve the manuscript.

Reviewer #2: The authors have presented their plan for a systematic review of parents' knowledge, competence, awareness, and experience of play with their children. This is an important and interesting topic that will benefit from their review.

I have some concerns regarding the current plan that I think need to be reconsidered.

- The age range of the children is too broad. Play between parents and infants/toddlers/pre-schoolers/children/teenagers are all wildly different. Including all ages will confound the results and make interpretations difficult. I would recommend reducing the age range.

- The primary outcomes are on important topics, but it is unclear how they will be operationalised, especially given that the secondary outcome is to identify if there are any measures that exist to measure these constructs.

- The planned year range to be searched is too broad. Parenting has changed dramatically even in the last 10 years. Including any study ever published, as appears to be the plan as outlined in the search strategy, will confound the results.

- The authors talk about parents throughout the proposal. Parenting, and especially play, has been shown to be very different for mothers and fathers. This needs to be taken into consideration when collecting and presenting the data and research findings.

- There are some minor grammatical errors that need revising.

7. PLOS authors have the option to publish the peer review history of their article (what does this mean?). If published, this will include your full peer review and any attached files.

Reviewer #1: No

Reviewer #2: No

---

## [Author Response · Author response to Decision Letter 0]

29 Jun 2022

Response to Reviewers

Editor comments:

Thank you for submitting your manuscript to PLOS ONE. After careful consideration, we feel that it has merit but does not fully meet PLOS ONE’s publication criteria as it currently stands. Therefore, we invite you to submit a revised version of the manuscript that addresses the points raised during the review process.

When revising your manuscript please ensure your address fully the reviewers' comments regarding the selection criteria, in particular those regarding the difference in play between infants and adolescents, and the timescale of the articles to be included given changes in parenting behavior over time.

Response:

We thank the editor for giving us an opportunity to revise and resubmit the manuscript. We have carefully read all the comments from the reviewers and addressed them appropriately. As per the suggestions, we have excluded the perceptions of play among parents of adolescents and to organize the findings from mothers of children in early and middle childhood separately. Likewise, we agree that parenting change over time. If any such trend will be observed in parents’ awareness, knowledge, and experiences of play, it will be reported in our results.

Journal Requirements:

Comment 1:

Response:

We have adhered to PLOS ONE’s styles to the best of our understanding

Comment 2:

Thank you for stating the following in your Competing Interests section: "NO authors have competing interests"

Response:

We have added the statement in the cover letter as advised.

Comment 3:

In your Data Availability statement, you have not specified where the minimal data set underlying the results described in your manuscript can be found. PLOS defines a study's minimal data set as the underlying data used to reach the conclusions drawn in the manuscript and any additional data required to replicate the reported study findings in their entirety. All PLOS journals require that the minimal data set be made fully available. For more information about our data policy, please see http://journals.plos.org/plosone/s/data-availability.

Response:

This is a systematic review protocol. Hence there is no data generated yet. However, we have attached the search criteria and data extraction forms as a supporting information file (S2 Additional File - Search Strategy. S3 Additional File - Data Extraction Forms)

Comment 4:

Please note that in order to use the direct billing option the corresponding author must be affiliated with the chosen institute. Please either amend your manuscript to change the affiliation or corresponding author, or email us at plosone@plos.org with a request to remove this option.

Response:

The corresponding author, Brightlin Nithis Dhas is still affiliated with Hamad Medical Corporation. The corresponding email has been changed to the institutional email (bdhas@hamad.qa) in the manuscript.

Comment 5:

Thank you for stating the following in the Financial Disclosure section: "The authors received no specific funding for this work."

We note that one or more of the authors are employed by a commercial company: Hamad Medical Corporation

Response:

We have added the statement in the cover letter as advised and in the declaration of interests section of the manuscript.

Comment 6:

Please include captions for your Supporting Information files at the end of your manuscript, and update any in-text citations to match accordingly. Please see our Supporting Information guidelines for more information: http://journals.plos.org/plosone/s/supporting-information. 

Response:

We have added the captions of the supporting information files at the end of the manuscript (lines 262 to 265).

Comment 7. 

We note that this manuscript is a systematic review or meta-analysis; our author guidelines therefore require that you use PRISMA guidance to help improve reporting quality of this type of study. Please upload copies of the completed PRISMA checklist as Supporting Information with a file name “PRISMA checklist”.

Response:

This is a systematic review protocol. The PRISMA-P checklist is added as an additional file. 

Reviewer Comments to the Author

1. Does the manuscript provide a valid rationale for the proposed study, with clearly identified and justified research questions?

Reviewer #1: Yes

Reviewer #2: Yes

Response:

We thank the reviewers for the positive comments

2. Is the protocol technically sound and planned in a manner that will lead to a meaningful outcome and allow testing the stated hypotheses?

Reviewer #1: Partly

Reviewer #2: Partly

Response:

We have addressed each of the comments and suggestions proposed by the reviewers. Our responses to each of the comments and the reference to the corrections made by us are described under each specific comment.

3. Is the methodology feasible and described in sufficient detail to allow the work to be replicable?

Reviewer #1: Yes

Reviewer #2: Yes

Response:

We thank the reviewers for the positive comments

4. Have the authors described where all data underlying the findings will be made available when the study is complete?

Reviewer #1: No

Reviewer #2: No

Response:

This is a systematic review protocol. The PRISMA-P checklist is added as S1 additional file. Once the study is complete, the PRISMA checklist will be published with the report. Data extraction forms and risk of bias assessment forms will be made available as supplementary materials. This is mentioned in lines 276 to 279 in the manuscript

5. Is the manuscript presented in an intelligible fashion and written in standard English?

Reviewer #1: Yes

Reviewer #2: Yes

Response:

We thank the reviewers for the positive comments

Reviewers' comments:

Reviewer 1

Comment 1:

The manuscript presents a registered report protocol for a systematic review on parents’ awareness, knowledge, competence and experiences of play. In addition, the authors aim to compare these between parents of children with disabilities and parents of typically developing children. The proposed study is original and should be of interest for the field. Although it is widely accepted that play has a major role in human development, the influence of parental variables has often been neglected in this research field.

Response:

We thank the reviewer for their positive comment on our work.

Comment 2.

I have a few general comments for the authors that I feel should be addressed before the manuscript is further considered for peer review:

Introduction

In general, although very briefly described, the theoretical background presented does point well towards the main research question. However, I think that some more general descriptions of the main concepts would be helpful to lay the foundation for the Methods section. For example, it remains unclear what the authors refer to as “play” and which developmental stage the authors refer to. It is quite obvious that the quality and quantity of children’s play (and thus also parents attitudes towards and their roles in it) will change dramatically throughout development. This is has to be described and the authors have to find a way to account for these developmental transitions in their study, either by focusing on a specific developmental stage or by taking “age” into account as a study variable. Furthermore, the introduction mentions parent’s awareness, knowledge, and competence as the main outcome variables. However, in the methods section (and abstract) it seems that four variables are included: awareness, knowledge, competence and experience. The title, on the other hand, only mentions “awareness, knowledge, and experiences”. This should be resolved and the introduction should mention and explain the main study variables in more detail. The introduction also lacks a particular motivation for the comparison between parents of children with disabilities (again: what do the authors mean here?) and parents of typically developing children.

Response:

We thank the reviewer for the suggestion to add general descriptions to the main concepts in the introduction. The discrepancies in parent and professional understanding of play is added to the introduction (lines 52 to 60). More information on what type of play is considered in the review is provided in the Methods section (lines 153 to 156). As suggested, more descriptions of the outcome variables, awareness, knowledge, and experiences are added in introduction (lines 63 to 65, and 85 to 87) and in Methods section (lines 168 to 172). We welcome the suggestion to take age into account as a study variable. We have included this and in addition to age, we have planned to explore variations in gender of parents, culture, and disability status of the child and report it appropriately (lines 90 to 97). We would like to thank the reviewer for pointing out the discrepancies in title, introduction, and methods sections of the manuscript on outcome variables. We have corrected this in the revised version of this manuscript.

Comment 2:

Methodology

The description of participants states that parents with “children” above 18 years of age will be excluded. Despite the fact that a 17-year-old is barely referred to as a “child” (but rather as an adolescent), the theoretical background presented seems to be focused on children’s play, and not of adolescents. Thus, the authors have to explain why studies with adolescents should be included in the literature review and, if so, have to adapt the theoretical background in order to capture the developmental transition of play from infancy to adulthood.

Response:

We thank the reviewers for this suggestion. We have changed the criteria in lines 138 and 139 to exclude studies reporting parents of adolescents (above 11 years).

Comment 3: 

Some more minor comments:

1. The types of studies included does not exclude previous reviews or meta-analyses – how do the authors go about these studies?

2. What is the reason to search in five different databases and to have three different independent reviewers?

3. The authors state as a limitation that “...since there are numerous terminologies used to describe play and parents’ understanding of it some relevant studies could potentially be missed”. This is why it is important that the authors state their conceptualization of “play” in the introduction and explain in the Methods section how they apply this definition in the literature search.

I hope that the above suggestions help the authors to improve the manuscript.

Response:

1. It was an omission from our part in our exclusion criteria to exclude previous reviews or meta-analyses but plan to review studies that have been included within those reviews if suitable using our inclusion/exclusion criteria. We have clarified this in lines 145 to 148.

2. Since this is a systematic review, we included five different databases to make it as comprehensive as possible. Three independent reviewers are part of this study and their work will help to reduce bias in selection of articles

3. We have added our definition of play to select articles in Methods section (lines 153 to 156) and described the discrepancies between parent and professional understanding of play in the introduction (lines 52 to 60). In addition, as an outcome of our systematic review, we will compile a list of definitions of play mentioned by parents.

The reviewer suggestions have definitely helped to improve the quality of our manuscript and we thank the reviewer.

Reviewer 2: 

Comment 1:

The authors have presented their plan for a systematic review of parents' knowledge, competence, awareness, and experience of play with their children. This is an important and interesting topic that will benefit from their review.

Response:

We are encouraged to see the positive comments from the reviewer and thank them for the same

Comment 2:

I have some concerns regarding the current plan that I think need to be reconsidered.

- The age range of the children is too broad. Play between parents and infants/toddlers/pre-schoolers/children/teenagers are all wildly different. Including all ages will confound the results and make interpretations difficult. I would recommend reducing the age range.

Response:

We thank the reviewer for their comments. We have reduced the age range from birth to 11 years. We will organize the results based on studies reporting the following developmental stages: Early childhood (0-5 years) and middle childhood (6-11 years). Depending on the number of studies within these age groups, we will further divide early childhood into infancy, toddlers, and preschoolers.

Comment 3:

- The primary outcomes are on important topics, but it is unclear how they will be operationalised, especially given that the secondary outcome is to identify if there are any measures that exist to measure these constructs.

Response:

We thank the reviewer for seeking this clarification. We have added more clear operational definitions for the outcome variables (lines 167 to 172)

Comment 4

- The planned year range to be searched is too broad. Parenting has changed dramatically even in the last 10 years. Including any study ever published, as appears to be the plan as outlined in the search strategy, will confound the results.

Response:

We agree with the reviewer that parenting attitudes about play has changed over the years. However, we want to identify this as part of the review. If any such trend are observed in parents’ awareness, knowledge, and experiences of play, it will be reported in our results.

Comment 5:

- The authors talk about parents throughout the proposal. Parenting, and especially play, has been shown to be very different for mothers and fathers. This needs to be taken into consideration when collecting and presenting the data and research findings.

Response:

We agree with the reviewer on the difference in perceptions between fathers and mothers on play. We will take this into consideration and present the findings appropriately in the results. We thank the reviewer for suggesting this.

Comment 6:

- There are some minor grammatical errors that need revising.

Response

We have corrected the grammatical errors and have provided a clean version of the manuscript alongside one with tracked changes.

General:

In addition to editorial and reviewer recommended changes, we have added additional references to any changes made and corrected grammatical errors.

---

## [Decision Letter · Decision Letter 1]

25 Aug 2022

Parents’ awareness, knowledge, and experiences of play and its benefits in child development: a systematic review protocol

PONE-D-21-25685R1

Dear Dr. Dhas,

We’re pleased to inform you that your manuscript has been judged scientifically suitable for publication and will be formally accepted for publication once it meets all outstanding technical requirements.

Kind regards,

Emily Freeman

Guest Editor

PLOS ONE

Additional Editor Comments (optional):

I would like to acknowledge that I participated as a reviewer for the intitial evaluation of this manuscript. Both myself, and the additional reviewer were satisfied with the changes made in response to our intial reviews.

Reviewers' comments:

Reviewer's Responses to Questions

**Comments to the Author**

1. Does the manuscript provide a valid rationale for the proposed study, with clearly identified and justified research questions?

Reviewer #1: Yes

2. Is the protocol technically sound and planned in a manner that will lead to a meaningful outcome and allow testing the stated hypotheses?

Reviewer #1: Yes

3. Is the methodology feasible and described in sufficient detail to allow the work to be replicable?

Reviewer #1: Yes

4. Have the authors described where all data underlying the findings will be made available when the study is complete?

Reviewer #1: Yes

5. Is the manuscript presented in an intelligible fashion and written in standard English?

Reviewer #1: Yes

6. Review Comments to the Author

You may also provide optional suggestions and comments to authors that they might find helpful in planning their study.

Reviewer #1: The authors have done a great job addressing my comments. I would therefore recommend to accept the manuscript for publication.

7. PLOS authors have the option to publish the peer review history of their article (what does this mean?). If published, this will include your full peer review and any attached files.

Reviewer #1: No

---

## [Editor Report · Acceptance letter]

30 Aug 2022

PONE-D-21-25685R1 

Parents’ awareness, knowledge, and experiences of play and its benefits in child development: a systematic review protocol 

Dear Dr. Dhas:

I'm pleased to inform you that your manuscript has been deemed suitable for publication in PLOS ONE. Congratulations! Your manuscript is now with our production department. 

Kind regards, 

on behalf of

Dr. Emily Freeman 

Guest Editor

PLOS ONE